# ScissorBot: Learning Generalizable Scissor Skill for Paper Cutting via Simulation, Imitation, and Sim2Real

**Jiangran Lyu**[1,2], **Yuxing Chen**[1,2], **Tao Du**[3], **Feng Zhu**[2], **Huiquan Liu**[2], **Yizhou Wang**[1,†], **He Wang**[1,2,†]

[1]CFCS, School of Computer Science, Peking University   [2]Galbot   [3]IIIS, Tsinghua University

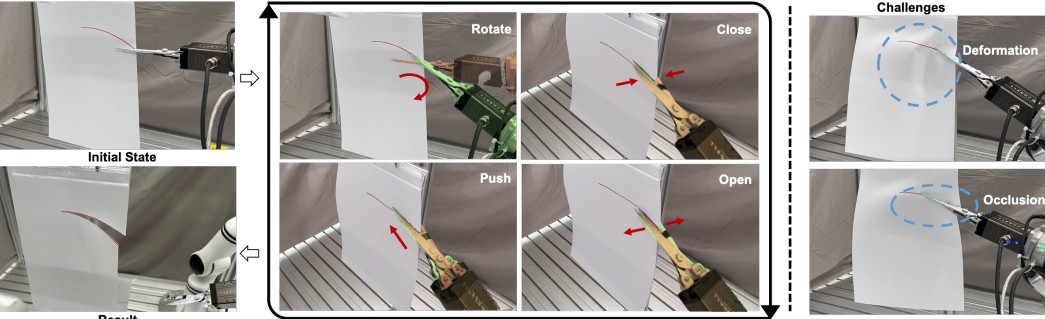

Figure 1: **Robotic paper cutting with scissors. Left:** The objective is to drive scissors to accurately cut curves drawn on the paper, which is hung with the top edge fixed. **Middle:** Our execution follows an action primitive sequence, namely *Rotate*, *Close*, *Open*, *Push*. The meticulous action, visualized as scissors before (orange) and after (green) each action, ensures accurate cutting in the real world. **Right:** During execution, large deformation of paper and severe occlusion between scissors and target curves occasionally occurs. Project Page: https://pku-epic.github.io/ScissorBot/

**Abstract:**

This paper tackles the challenging robotic task of generalizable paper cutting using scissors. In this task, scissors attached to a robot arm are driven to accurately cut curves drawn on the paper, which is hung with the top edge fixed. Due to the frequent paper-scissor contact and consequent fracture, the paper features continual deformation and changing topology, which is diffult for accurate modeling. To deal with such versatile scenarios, we propose ScissorBot, the first learning-based system for robotic paper cutting with scissors via simulation, imitation learning and sim2real. Given the lack of sufficient data for this task, we build PaperCutting-Sim, a paper simulator supporting interactive fracture coupling with scissors, enabling demonstration generation with a heuristic-based oracle policy. To ensure effective execution, we customize an action primitive sequence for imitation learning to constrain its action space, thus alleviating potential compounding errors. Finally, by integrating sim-to-real techniques to bridge the gap between simulation and reality, our policy can be effectively deployed on the real robot. Experimental results demonstrate that our method surpasses all baselines in both simulation and real-world benchmarks and achieves performance comparable to human operation with a single hand under the same conditions.

**Keywords:** Deformable Object Manipulation, Imitation Learning, Sim-to-Real

## 1   Introduction

Paper cutting, an ancient craft dating back to at least the 6th century [1], has evolved alongside human civilization, serving as a medium for emotional and symbolic expression [2]. In modern society,

---

† Corresponding Authors

8th Conference on Robot Learning (CoRL 2024), Munich, Germany.

it has wide applications ranging from decorative art [3] and education to advanced manufacturing and technology [4, 5]. Humans can use scissors to perform paper cutting, showcasing their dexterity in tool usage. However, robots have yet to master this generalizable cutting skill, which indicates using scissors to cut painted patterns on paper with visual observation. The main obstacle lies in the intricate interaction between paper and scissors, characterized by the continual deformation and changing topology of the paper. Accurately modeling this dynamics based on first principles and achieving precise control using Linear Quadratic Regulator or Model Predictive Control is highly challenging. In contrast, learning-based methods, benefiting from data-driven, offer a promising alternative without the need for explicit modeling of this complex system and have the potential to achieve generalization across diverse cutting tasks.

Despite the strengths of learning-based approaches for many robotic tasks [6, 7, 8, 9, 10, 11], they fall short on this task partly due to the following challenges. First, there is insufficient data for paper-cutting tasks so far. Collecting accurate scissor-cutting demonstrations directly on a real-world robot system is laborious and hazardous, making simulation essential for data generation. However, existing simulators for thin-shell objects [12, 13, 14] do not support detailed interaction with scissors including contact and consequent fracture. Second, learning a generalizable and accurate scissor policy is intrinsically difficult for this paper cutting task. Given its contact-rich and deformable nature, even millimetric movements of the scissors can induce significant bending (as illustrated in Fig. 1) or lead to curves deviating substantially from the intended target. Meanwhile, the scissors occasionally occlude the target curve, leading to an ill-posed decision-making problem. In this scenario, reinforcement learning methods struggle with poor data efficiency. Third, the significant sim-to-real gap, both physical and visual, hinders the deployment of the policy in the real world. The physical gap, arising from the complex factors in the real world, sometimes leads to deviations in the cutting trajectory during execution. Additionally, the visual gap for this task is mainly due to the *edge bleeding artifact* [15], where real-world sensor depth blurs at object edges. This blurring causes jittery observations of the scissor-paper interaction, leading to incorrect policy decisions.

To address the above challenges, we introduce ScissorBot, the first learning-based robotic system for paper cutting with scissors via a combination of simulation, imitation learning and sim2real techniques. To mitigate the data scarcity issue, we develop PaperCutting-Sim, a paper-cutting simulator supporting interactive fracture coupling with scissors, enabling large-scale demonstration generation with a heuristic-based oracle policy. This oracle policy leverages privileged information that is inaccessible in the real world, allowing us to distill its knowledge into a vision-based imitation learning policy. To handle tasks of significant complexity with improved efficiency, we customize an action primitive sequence which constrains the action space of learning and allivating potential compounding errors. We also use multi-frame point clouds as input to complete the occluded and underlying dynamics information. To bridge the physical and visual gaps between simulation and reality, we propose data augmentation on deviation correction and artifact mimicry, respectively. The former method can adaptively correct compounding errors via adding out-of-distribution data which pairs deviation states with corresponding correction action. The latter aims to mimic edge bleeding artifact in the simulation to achieve visual alignment with reality.

Through extensive simulated and real-world experiments, we evaluate the efficacy and generalizability of our learning policy across three different task difficulty levels: *Easy*, *Middle*, and *Hard*. Our method improves cutting accuracy by at least fivefold compared to the best alternative methods, as measured by Chamfer distance. Furthermore, despite training only on *Easy* data from simulation, our method achieves a Chamfer distance of 2mm and an IoU of 89 for *Middle* and *Hard* patterns in the real world, a performance that is comparable to humans with single-hand operation. Our research opens up new oppotunities for contact-rich and fine manipulation of deformable objects.

## 2 Related Work

### 2.1 Deformable Object Manipulation

The manipulation of deformable objects, such as dough [16, 17], cloth [18, 8, 19] and rope [20, 13] has been extensively studied in the in the scentific and engineering disciplines. Zhao et al. [21]

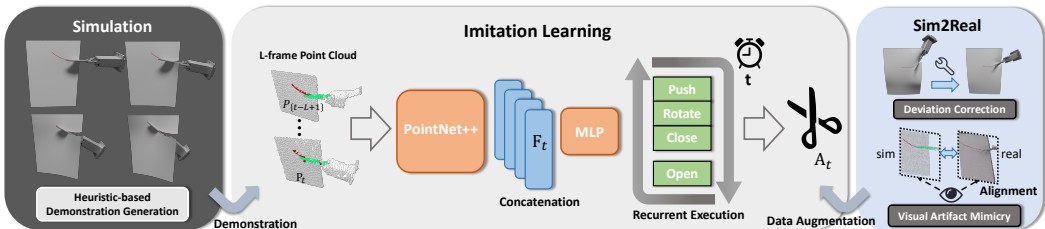

Figure 2: **An overview of the learning system.** The system first generates expert demonstrations in our built simulation which supports interactive fracture of the paper. These demonstrations are then used to train a vision-based imitation learning policy that inputs multi-frame point clouds (Blade point cloud is highlighted in green only for visualization) and outputs parameters of action primitive. Meanwhile, Deviation Correction and Visual Artifact Mimicry provide data augmentation to imitation learning which ensures a robust transfer from simulation to real world.

train robots in the real world to learn paper-flipping skills, and Namiki et al. [22] explore paper folding using motion primitives. Other studies focus on kirigami [23, 24], the traditional art of folding and cutting paper to create intricate designs. For paper cutting, these studies typically use desktop cutting plotters rather than fully automated robotic systems. Additionally, various robotic systems have been developed for cutting deformable objects in different domains, such as vegetables [25, 26], dough [16, 17], and soft objects with rigid cores [27]. However, these systems generally employ tabletop knife cutting, which differs from our approach of using scissors for cutting.

## 2.2 Simulation Environments for Paper Cutting

One line of works build simulators to boost robotic skill learning for thin-shell materials [13, 14, 28, 12, 29], however they couldn't simulate the fracture during the scissors cutting. Some works focus on simulating the cutting process of soft materials [30, 27]. Other works studies paper fracture process either from the theoretical analysis [31, 32], re-meshing algorithm [33, 34] or its application in kirigami [35, 36]. Overall, none of the existing works implements the paper cutting simulation for robot learning, which combines dynamic modeling of paper and interactive fracture interweaving paper remeshing according to scissor motion.

## 2.3 Imitation Learning

Imitation Learning (IL) [37, 6, 7, 38] is a supervised learning methodology for training embodied agents using expert demonstrations. The commonly used Behavior Cloning (BC) [37] strategy directly trains the policy to imitate expert actions. Despite its simplicity, this approach has demonstrated remarkable effectiveness in robotic manipulation [39, 40]. In this paper, we adopt imitation learning and ultilze action primitive sequence to ensure robustness during execution.

# 3 Method

## 3.1 Task Formulation and Method Overview

The objective of this task is to accurately cut paper along drawn curves using scissors, guided by single-view point cloud input. The scissors are mounted on a single robotic arm, with the top edge of the paper fixed and the bottom edge free, enabling generalization to various scenarios.

In order to learn such generalizable scissor skills, we introduce ScissorBot, a robotic system designed to learn visuo-motor policies via simulation, imitation learning and sim2real techniques. As the system depicted in Fig. 2, we first present our training data source in Sec. 3.2, where we develop the first paper-cutting simulator, PaperCutting-Sim, and heuristic-based demonstrations generation in Sec. 3.3. We then detail the vision-based imitation learning design in Sec. 3.4 and sim2real techniques in Sec. 3.5. Finally, we present a hardware setup for deploying our policy in real-world scenarios in Sec. 3.6.

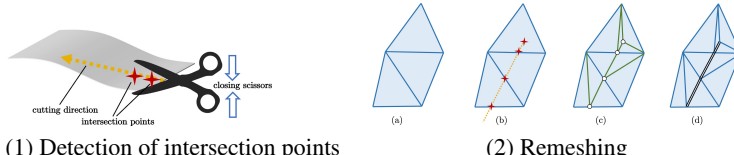

(1) Detection of intersection points          (2) Remeshing

Figure 3: **Interactive Fracture in our PaperCutting-Sim.** (1): As the scissors close, the fracture occurs along the cutting direction. Intersection points (**red star**) can be computed from edge-edge detection and vertex-face detection between the cutting direction (**orange dashed**) and the paper mesh. (2): (a) The original paper mesh (**blue triangles**). (b) Intersection point (**red star**) and cutting direction (**orange dashed**). (c) According to the intersection points, new vertices are added on the existing edges and the endpoint is inserted inside the triangle. The new edges (**green solid**) are connected between the new inserted vertex and the opposite vertex in the triangle. (d) The edges between these newly added vertices are split into two pieces (**black solid**).

## 3.2 PaperCutting-Sim

We build a paper-cutting simulator, PaperCutting-Sim, to support the modeling of both paper and scissors, as well as their contacts and the consequent fracture. The simulator is implemented in Python and Taichi [41], which supports parallel computation on GPUs.

**Dynamic Model.** Following the Kirchhoff-Love shell theory, we model the elastic energy of the paper as the sum of stretching elastic energy and bending elastic energy. The stretching elastic energy is modeled as co-rotational linear elasticity, and the bending elastic energy is calculated using squared difference of mean curvature [42]. We perform spatial discretization using the Finite Element Method [33], and update positions and velocities through implicit time integration [43], which optimizes the incremental potential via Newton's method. To model the contact between the paper and the scissors, we represent the scissors using a signed distance field, utilize the cubic of the signed distance to calculate collision energy, and apply Coulomb friction.

**Interactive Fracture.** Different from cutting simulation with predefined fracture surfaces [30], interactively handling fracture coupled with contact is a non-trivial problem. To address this, we design a two-phase geometry-based approach, as illustrated in Fig. 3. First, during the closing process of the scissors, we propose using edge-edge detection and vertex-face detection to detect the intersection points between the cutting trajectory and the paper mesh. Then these intersection points are added to the paper mesh and related edges are connected and split according to the vertex position relationship inside triangles. We refer the reader to Appendix B for more details.

## 3.3 Demonstration Generation

In this section, we devise an action primitive sequence and a heuristic-based Oracle policy for large-scale demonstration generation. For distillation, we preserve high-quality demonstrations measured by chamfer distance.

**Action Primitive Sequence.** We designed four action primitives for the scissors in a heuristic manner, namely *Open*, *Push*, *Rotate*, and *Close*. These primitives can be combined into a sequence to cut a straight line on the paper as follows: (1) Open the scissors to the maximum extent. (2) Push the scissors to the starting point of the line. (3) Rotate the scissors towards the endpoint of the line. (4) Close the scissors breaking the paper. Note that the pushing action is a 1D translation along the cutting direction as we approximate that the starting point is in the scissor cut direction.

**Oracle Policy.** The oracle policy initially discretizes the target smooth curve into several line segments, with this approximation scarcely impacting visual appearance. Subsequently, the entire curve can be cut by multiple action sequences for line segments iteratively. As oracle policy can access the 3D position of target line during each step, the pushing distance ($p \in \mathbb{R}^1$), rotation matrix ($\mathbf{R} \in \mathrm{SO}(3)$), and closed angle ($c \in \mathbb{R}^1$) can be computed by relative position between scissors and target line. Please refer to Appendix C for more details.

## 3.4 Vision-based Imitation Learning

This section presents the design of our learning framework, with multi-frame point clouds as input and action parameters as ouput. The complete network architecture is depicted in Fig. 2.

**Spatial-Temporal Observation Encoding.** We first pre-process raw single-view point cloud using bounding-box cropping and FPS sampling. Then sequential $L$-frame point clouds $\{P_{t-i-1}\}_{i=1}^{L}$, along with a binary mask indicating whether a visible point originates from the target curve, are fed into a shared PointNet++ encoder [44] to obtain features $\{\mathcal{F}_{t-i+1}\}_{i=1}^{L}$. These features are concatenated and passed through a shallow MLP to regress actions parameters.

**Primitive Learning.** In contrast to selecting the direct 7 DoF scissor pose as our output action space, the output action parameters are associated with the designed action primitives mentioned in Sec. 3.3. These actions are recurrently executed for each stage which keeps the order of *Push*, *Rotate*, *Close*, *Open* repeatedly. We employ Mean Squared Error (MSE) loss for the *Push* and *Close* terms, and 9D L1 Loss [45] for *Rotate*. The overall loss is formulated as:

$$\mathcal{L} = \lambda_p(p - \hat{p})^2 + \lambda_c(c - \hat{c})^2 + \lambda_{\mathbf{R}} \sum_{i,j} |\mathbf{R}_{ij} - \hat{\mathbf{R}}_{ij}| \tag{1}$$

where $\lambda_p$, $\lambda_c$, $\lambda_R$ are respective weights, and $\hat{p}$, $\hat{c}$, $\hat{\mathbf{R}}$ are ground truth action values.

### 3.5 Sim-to-Real Transfer

**Deviation Correction.** We introduce deviation correction to enhance the robustness for drifting scenarios which sometimes occurs in the real world. In this approach, we fine-tune the trained model using correction data, which comprises out-of-distribution states paired with corrective actions. These data are generated by introducing random rotation perturbances to the action during oracle policy execution. As the oracle policy consistently cuts towards the endpoint of each line segment, the next action naturally corrects minor drifting errors.

**Visual Artifact Mimicry.** We propose a simple yet effective method to mimic *edge bleeding artifact* in the simulation. To create continuous value at the edge between foreground and background in simulated depth image, we preprocess the depth with an average pooling kernel and



Figure 4: **Visualization of Visual Artifact Mimicing** (a) Perfect point cloud in simulation with scissors blade highlighted (**green points**) (b) Point cloud with our proposed visual artifact mimicry. (c) Point cloud captured in the real world with artifact.

add random noise perpendicular to the surface of the paper to the point cloud of the blade. In this way, the artifact can be mimicked (Fig. 4(b)) in our training data thus reaching a visual alignment from simulation to reality.

### 3.6 Hardware System Design

We design a hardware system for the paper-cutting task, as Fig. 5 shown. The setup includes a Realman robot equipped with a scissor extension for manipulation and a single Kinect DK camera to capture RGBD observations. To secure the paper, we use plastic clips to fix the top edge, leaving the lower edge free. The target curves are drawn in red on white paper, with corresponding binary masks obtained through simple RGB-based segmentation. We use A4 printer paper ($210\,\text{mm} \times 297\,\text{mm}$, $75\,\text{g/m}^2$) as the material for the following experiments.

## 4 Experiments

In this section, we first evaluate the cutting performance of our proposed method through comparison with various baselines and variants in simulated environments. We further validate our approach in the real-world. Please refer to Appendix A for additional experiment information including generalization to different materials, ablation studies and more qualitative results.

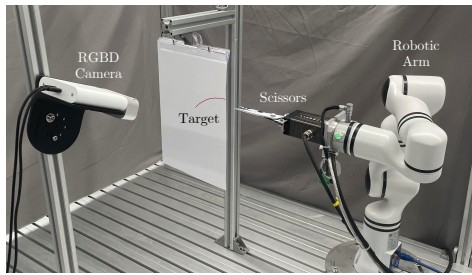

Figure 5: Hardware System

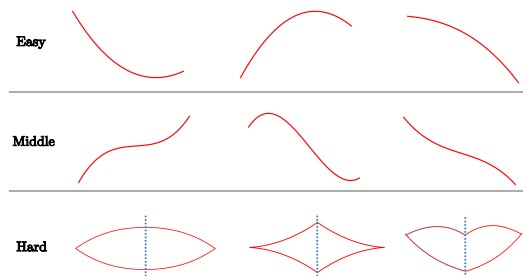

Figure 6: Example curves for Easy, Middle and Hard tracks.

## 4.1 Benchmark

**Task Datasets.** We focus on simple smooth curve cutting and split it into three distinct tracks, illustrated in Fig. 6. In each track, curves are generated using Bézier curves parameterized by four control points. By manipulating the positional relationship of these control points, we can control the second-order derivative of the curve, which in turn determines the complexity of the scissors' motion. The discussion on non-smooth and non-simple curves can be found in the Sec. 5.

- *Easy*: In this track, the second-order derivatives of curves are consistently positive or negative.
- *Middle*: Curves in this track exhibit varying positive and negative second-order derivatives.
- *Hard*: This track comprises several patterns, each composed of two curves from the *Easy* track. In real-world settings, this track can be further required to cut the origami sheet to obtain an axisymmetric closed-shape pattern.

To demonstrate the generalization capability of our policy, our training set consists of approximately 5k trajectories solely from the Easy track. There are 100 curves of each track for testing.

**Evaluation Metrics.** In our evaluation process, we utilize various metrics to gauge the quality of our results across different difficulty levels. For the all three tracks, we employ the chamfer distance as a measure of deviation between the cropped curve and the target curve. Additionally, we report the Recall metric under different thresholds of chamfer distance, indicating the proportion of well-cut instances. For trials completing closed shapes in the *Hard* track, we further assess the quality by calculating the mean Intersection over Union (mIoU) between the cropped pattern and the target pattern, providing a comprehensive measure of similarity and accuracy.

## 4.2 Policy Evaluation in Simulation

**Non-learning Baselines.**

- *Open-loop Planning* detects the target goal curve prior to cutting. Then it discretizes the detected curve into isometric line segments and plans the scissor translation and rotation at each step.
- *Online Fitting* employs an step-by-step line fitting utilizing RANSAC. The fitted line target from the captured point cloud determines the movement distance and scissor rotation at each step.

**Learning based Baselines**.

- *Direct Pose Regression*. This methodology directly regresses the 7 Degrees of Freedom (DoF) scissor pose (6D Pose and 1D joint angle) .
- *Action Chunking* [7]. In this policy, actions for the next $k$ timesteps are predicted. The current action to execute is determined from weighted averages across the previous overlapping action chunk. We adopt the implementation from [7].

**Comparison to Non-learning based Baselines.** Our method exhibits significantly superior cutting accuracy when compared to non-learning based approaches. As depicted in Fig. 7, the cutting trajectories of *Open-loop Planning* substantially deviates from the target curve. The reason lies in that *Open-loop Planning* lacks adaptability to environmental changes and accumulates a lot of errors during the highly non-linear interaction between scissors and paper. Although *Online Fitting*

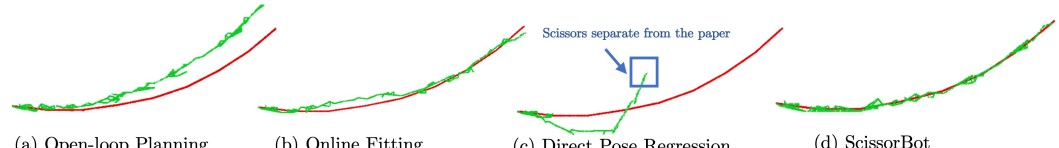

| | (a) Open-loop Planning | (b) Online Fitting | (c) Direct Pose Regression | (d) ScissorBot |

Figure 7: **Visualization of Cutting Results on UV plane of the paper.** Target curves are in red while cropped lines by scissors are in green.

| Methods | Easy | | | Middle | | | Hard | |
|---|---|---|---|---|---|---|---|---|
| | Chamfer (mm) | Recall@1.5 | Recall@5.0 | Chamfer (mm) | Recall@1.5 | Recall@5.0 | Chamfer (mm) | mIoU |
| Open-loop Planing | 10.8 | 9.3 | 25.0 | 6.8 | 10.5 | 36.3 | 18.1 | 31.2 |
| Online Fitting | 5.5 | 31.4 | 73.6 | 5.3 | 21.0 | 53.1 | 10.3 | 63.0 |
| **Ours** | **1.1** | **85.1** | **98.6** | **1.5** | **79.6** | **96.6** | **1.9** | **91.3** |
| Oracle | 1.4 | 83.1 | 98.2 | 1.4 | 80.1 | 98.2 | 1.9 | 92.2 |

Table 1: Comparison with non-learning based baselines in simulation.

demonstrates some adaptability by adjusting its cutting action based on current observations, it falls short of making optimal decisions, particularly in instances of occasional occlusion. In contrast, our learning policy showcases remarkable adaptability to dynamic conditions, resulting in notably precise operations, with a chamfer distance of $1.1\,\mathrm{mm}$ compared to $10.8\,\mathrm{mm}$ and $5.5\,\mathrm{mm}$ for *Open-loop Planning* and *Online Fitting* respectively.

**Effect of the Primitive Learning.** The comparison with learning-based baselines exhibits the effectiveness of our primitive learning. As shown in Fig. 7(c), the alternative frequently causes separation between scissors and paper and thus fails to perform the paper-cutting tasks to the end. We only report its chamber distance with the first one-third of the target curve in

| Method | Chamfer (mm) | Recall@1.5 | Recall@5.0 |
|---|---|---|---|
| Direct Pose Regression | 11.2* | 9.0* | 24.4* |
| Action Chunking [7] | 11.4* | 8.9* | 25.1* |
| Ours | 1.1 | 85.1 | 98.6 |
| Ours + Action Chunking [7] | 1.2 | 84.2 | 98.5 |

Table 2: Comparison with learning-based baselines in simulation. For *Direct pose regression* and its variance, we only calculate its chamber distance with the first one-third of the target curve (*).

Tab.2. Even incorporating *Action Chunking*, designed to mitigate compounding errors, there is no substantial improvement in completing the target curves. This is due to the drastic state transitions at each step, which complicate the accurate prediction of future actions. These results highlight the highly nonlinear nature of the task. In contrast, our designed primitive constraints the action space of scissors, which minimizes possible errors during execution.

**Generalization to Novel Curves and Patterns.** Middle and Hard targets pose additional challenges due to more varied patterns, leading to more complex deformation and fratcure during cutting. Despite being trained on the *Easy* track of curves, our policy demonstrates robustness and achieves performance comparable to the oracle policy when handling *Middle* and *Hard* targets. This success is due to its generalizability from careful system design.

### 4.3  Policy Evaluation in the Real World

We evaluate the performance of our sim2real model on a real-world platform. Quantitative and qualitative results are presented in Table 3 and Fig. 8, Fig. 11, respectively.

**Experiment Details.** To understand the capability of our system, we conduct a comprehensive user study to gather statistics on human performance. Specifically, we invite 10 subjects aged between 10 and 60 years, including both males and females. They are asked to cut paper using a daily scissor with one hand while the paper was hanging under the same conditions as our robot system. Each subject complete ten trials for each track.

We define "finished" as the cut line having a chamfer distance of less than 3 cm from the target curve, with no tearing or folding of the paper. After each trial for both policy and human, we capture the cropped paper using an RGBD camera and compute metrics such as chamfer distance and IoU. We report the median and extremum of these metrics in the form of $x \pm u$.

| Methods | Easy | | Middle | | Hard | |
|---|---|---|---|---|---|---|
| | Finished Rate | Chamfer (mm) | Finished Rate | Chamfer (mm) | Finished Rate | IoU |
| Human | 10/10 | $2 \pm 1$ | 10/10 | $2 \pm 1$ | 10/10 | $92 \pm 3$ |
| Ours w/o Visual Artifact Mimicry | 0/10 | - | 0/10 | - | 0/10 | - |
| Ours w/o Deviation Correction | 7/10 | $\mathbf{2 \pm 1}$ | 7/10 | $3 \pm 1$ | 7/10 | $84 \pm 8$ |
| Ours | **9/10** | $\mathbf{2 \pm 1}$ | **8/10** | $\mathbf{2 \pm 1}$ | **8/10** | $\mathbf{89 \pm 5}$ |

Table 3: Quantitative results in the real world.

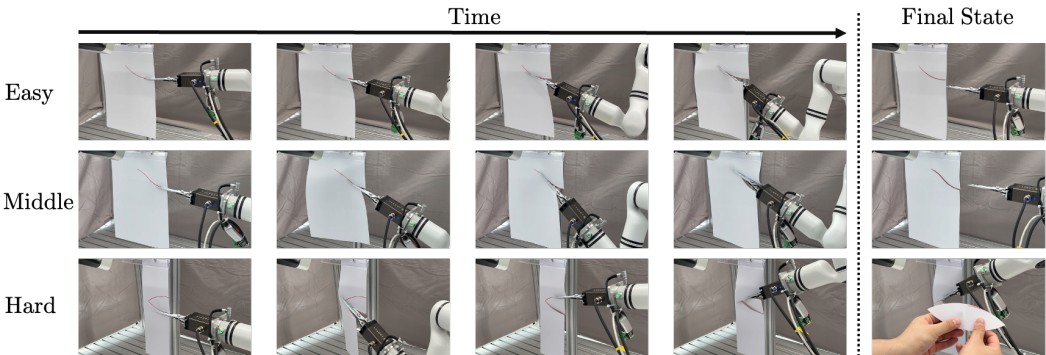

Figure 8: Realworld cutting process on Easy, Middle and Hard tracks.

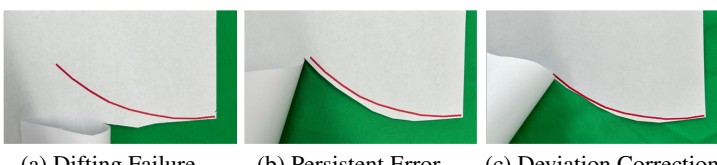

(a) Difting Failure     (b) Persistent Error     (c) Deviation Correction

Figure 9: **Qualitative result of our Deviation Correction.** Without using Deviation Correction, policy trained from simulation frequently falls into drifting failures (a) or keeps persistent errors (b). Our Deviation Correction (c) leads to a accurate and robust deployment.

**Result Analysis.** As evidenced in Table 3, our policy consistently fails without any sim2real, as a result of confused perception with the interactions between blades and paper. Our mimicry strategy mitigates the visual gap and minimize erratic action prediction, thereby achieve successful deployment. However, the performance is still still unsatisfactory. As illustrated in Fig. 9, policies sometimes experience drifting failures (Fig. 9a) or persistently exhibit errors (Fig. 9b). To this end, the correction mechanism addaptively corrects the deviation and enhances the stability and accuracy of the deployed policy. For example, it enhances the finished rate from 7/10 to 9/10 in the "Easy" track and reducing deviation by $1\,\mathrm{mm}$ in the Middle track. Combining the above sim2real techniques, our system achieves comparable performance to human single-hand manipulation under same condition, which has only $2\,\mathrm{mm}$ error from the target curve. Furthermore, it achieves an IoU of 89 on the *Hard* track, which is particularly challenging due to more drastic bending.

## 5 Conclusion, Limitations and Future Directions

We introduce ScissorBot, the first learning-based robotic system for generalizable paper cutting using scissors. The system utilizes demonstrations collected in our newly developed paper-cutting simulator to train a primitive-based imitation learning policy and combines sim2real techniques to achieve robust deployment in the real world. Extensive experiments exhibit the generalizability and accuracy of our system on simple smooth curves which cover most of cutting scenarios. However, scissor cutting for non-simple or non-smooth curves is still an open problem. Non-smooth curves like zig-zag patterns must result in the sudden relative pose change frequently for scissors while non-simple curves will result in a circle in the scissor trajectory. Both two kinds of scissors' motion can't be achieved considering our single-arm workspace. To solve these challenging situations, it may be a dexterous dual-arm cooperation system containing one arm holding and rotating the paper and the other manipulating scissors. We leave this bimanual hardware system with policy design as future works.

**Acknowledgments**

We would like to express our deepest gratitude to Ruihai Wu for his invaluable guidance during the early stages of this work, which was crucial to its success. We also thank Jiayi Chen for providing many valuable discussions, offering insightful feedback, and pointing out several key issues and potential improvements. We are grateful to Mi Yan, Jiazhao Zhang, Songlin Wei, Wenbo Cui, and Weikang Wan for their constructive feedback on the writing, figures, and tables in this paper. We also appreciate the help of Xiaomeng Fang and Hao Shen with the camera calibration, as well as the support and guidance of Jilong Wang on the robot workspace and motion planning. Additionally, we acknowledge Jiqiao Li, Mingdong Lu, Xinzhe Jin, Yuanchen Cao, Zhiqiang Xu, and Chenxu Zhao for their care and encouragement throughout this work.

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

# A More Experiments and Results

## A.1 Generalization to Different Materials

We highlight experiments that demonstrate the generalization to different materials. Specifically, we select six different materials for evaluation: cardboard, A3 printer paper, rice paper, plastic sheet, photo fabric, and aluminum foil. These materials exhibit varying physical properties, such as thickness, density, stiffness, and their bending and stretching characteristics. For example, cardboard is verythick and hardly bends during cutting, while rice paper is very thin and deforms significantly. A3 printer paper is larger in size than the A4 paper used in our previous experiments, and foil has an uneven surface. We conducted 10 trials for each material using the Easy track of curves.

By applying domain randomization to physical parameters of the paper including size, mass and Young's modulus, our policy successfully generalizes to these different materials without requiring customized training. We exhibit that learning to cut papers with scissors is more generalizable to different materials than specialized cutting machines. Videos can be found on our website.

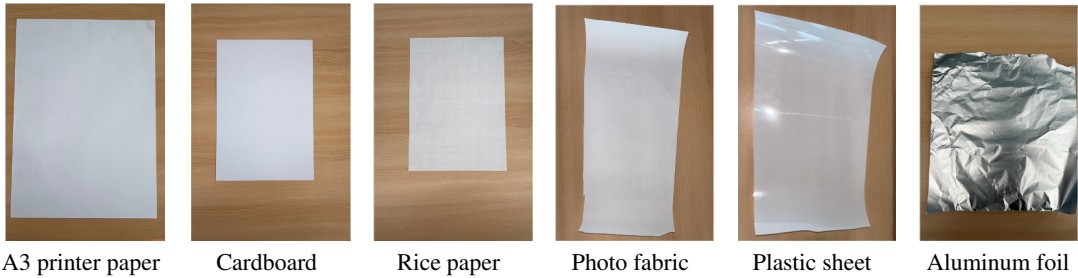

| A3 printer paper | Cardboard | Rice paper | Photo fabric | Plastic sheet | Aluminum foil |

Figure 10: Test on unseen materials.

| Metric | Cardboard | A3 printer paper | Rice paper | Photo fabric | Plastic sheet | Aluminum foil |
|---|---|---|---|---|---|---|
| Finish rate | 7/10 | 9/10 | 9/10 | 8/10 | 8/10 | 8/10 |
| Chamfer | 3±1 | 2±1 | 2±1 | 2±1 | 2±1 | 3±1 |

Table 4: Experiments on different materials.

## A.2 More Qualatative Results

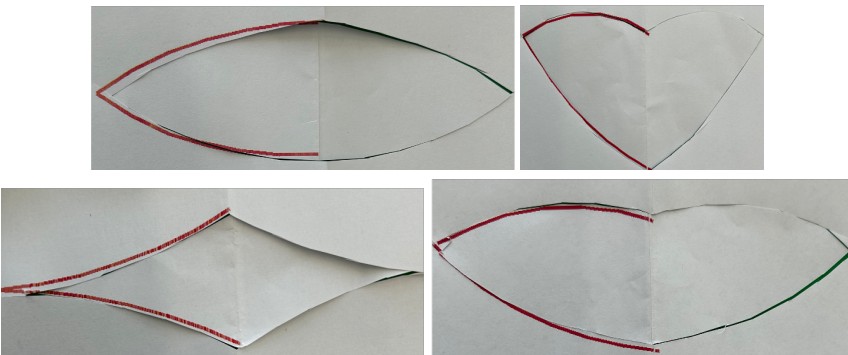

Figure 11: **Realworld results on patterns from Hard tracks.** The cropped pattern from our system has a accurate overlap with the target pattern.

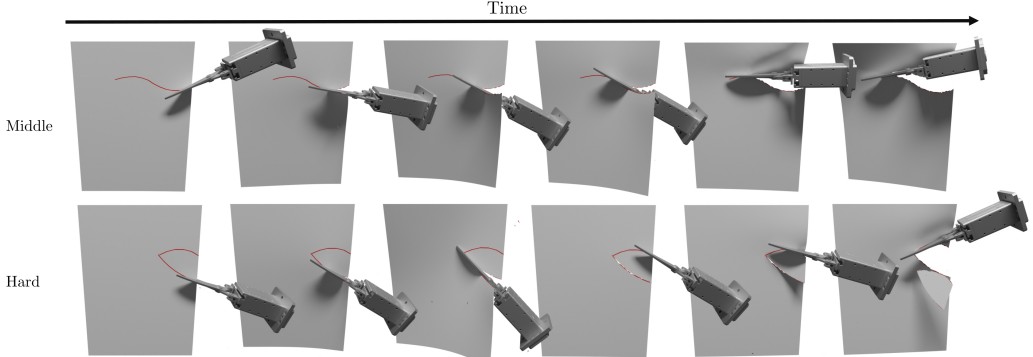

Figure 12: Cutting process of our method in the simulation on Middle and Hard tracks.

| Basic length (mm) | Steps | Chamfer (mm) |
|---|---|---|
| 20 | $\sim 29$ | 1.9 |
| 15 | $\sim 37$ | 1.4 |
| 10 | $\sim 53$ | 1.3 |

Table 5: Effects of Discretization Granularity for curves, i.e, the basic length of line segments.

### A.3 Ablation Studies

In this section, we conduct experiments to analyze the effects of critical parameters and design choices.

**Discretization Granularity.** During the generation of expert data, smooth curves are discretized into line segments as an approximation. We aim to study the impact of discretization granularity on efficiency and quality, as reported in Table 5. It's evident that as the length of each line segment decreases, the number of execution steps increases while the quality improves. When the basic length decreases from $15\,\mathrm{mm}$ to $10\,\mathrm{mm}$, the reduction in chamfer distance is marginal, merely $0.1\,\mathrm{mm}$. Thus, we opt for a trade-off choice of $15\,\mathrm{mm}$, with a balance between efficiency and quality.

**Observation Horizon** ($L$)**.** We analyze the effect of different observation horizons, as detailed in Table 6. Policies with 4 or 6 frame horizons outperform that without temporal inputs by approximately 10 points on Recall@1.5. The superiority of our spatial-temporal encoding lies in its robustness to occlusion and deformable dynamics. Although increasing the horizon from 4 to 6 yields marginal improvements in cutting quality, e.g., 0.3 for Recall@1.5, we opt for a horizon of 4 in our system to maintain a favorable balance between performance and efficiency.

**Filtering Threshold** ($\tau$)**.** We train our imitation policy using various demonstration filtering thresholds ranging from 0.7 to 1.6, as well as no filtering, as summarized in Table 7. A stricter threshold yields better performance and a lower data usage rate, necessitating the generation of more demonstrations for distillation. In our system, we select $\tau = 1.0$ to strike a balance between data quality and efficiency.

## B    Details of PaperCutting-Sim

We consider the occurrence of paper fractures due to the intersection of the scissor blades. As the scissors close, the intersection point of the two blades, denoted as $\mathbf{P}$, moves along the paper surface. The trajectory of this movement forms the fracture line on the paper. At each time step, we detect this moving trajectory on the paper mesh and perform remeshing, which includes vertex insertion, edge connection, and edge splitting.

**New Vertex Insertion:** For each time step $t$, the movement of $\mathbf{P}$ can be represented as a segment $E = (\mathbf{P}_t, \mathbf{P}_t - \alpha \mathbf{V}_t)$, where $\alpha$ is the velocity and $\mathbf{V}$ is the cutting direction of the scissors. Per-

| Horizon | Chamfer(mm) | Recall@1.5 | Recall@5.0 |
|---|---|---|---|
| w/o multi-frame | 1.5 | 74.4 | 94.2 |
| $L = 2$ | 1.6 | 70.4 | 93.1 |
| $L = 4$ | 1.1 | 85.1 | 98.6 |
| $L = 6$ | 1.1 | 85.4 | 98.9 |

Table 6: Ablation study of Temporal encoding and Observation Horizon.

| Filter | Data usage | Chamfer(mm) | Recall@1.5 | Recall@5.0 |
|---|---|---|---|---|
| w/o filtering | 100% | 1.7 | 74.5 | 92.5 |
| $\tau = 1.6$ | $\sim 90\%$ | 1.4 | 81.8 | 97.8 |
| $\tau = 1.0$ | $\sim 70\%$ | 1.1 | 85.1 | 98.6 |
| $\tau = 0.7$ | $\sim 30\%$ | 1.1 | 85.0 | 98.6 |

Table 7: Ablation study on the Filtering threshold $\tau$.

forming edge-edge detection between $E$ and the paper mesh $M$ will yield most intersection points. For the endpoints of $E$, they may not be located at an existing edge, so we also perform vertex-face detection to get the intersection points. We insert all the intersection points as new vertices in order.

**New Edge Connection.** With the insertion of vertices, we perform edge connection. For a newly inserted vertex on an existing edge, we connect it to the opposite vertices in all triangles containing that edge. For a newly inserted vertex inside a triangle but not on the edge, we connect it with three vertices of the triangle.

**Edge Splitting:** For each newly connected edge, if its two endpoints are newly inserted vertices, then this edge need to be splitter.

## C   Details of Demonstration Generation

The action parameters are computed through relative pose between scissors (blade intersection $P_t$ and cutting direction $\mathbf{V}_t$) and target line segment ($\mathbf{Tar}_t^k = (s_t^k, s_t^{k+1})$) for each step $t$.

When $t$ is the step for *Push*, the pushing distance $p_t$ is computed as :

$$p_t = \frac{(s_t^k - P_t) \cdot \mathbf{V}_t}{\|\mathbf{V}_t\|^2} \mathbf{V}_t \tag{2}$$

When $t$ is the step for *Rotate*, the Rotation $\mathbf{R}_t$ is computed using Rodrigues' rotation formula:

$$\mathbf{w} = \mathbf{V}_t \times \mathbf{Tar}_t^k \tag{3}$$

$$\theta = \cos^{-1}(\mathbf{V}_t \cdot \mathbf{Tar}_t^k) \tag{4}$$

$$\mathbf{K} = \begin{bmatrix} 0 & -w_z & w_y \\ w_z & 0 & -w_x \\ -w_y & w_x & 0 \end{bmatrix} \tag{5}$$

$$\mathbf{R}_t = \mathbf{I} + \sin\theta\mathbf{K} + (1 - \cos\theta)\mathbf{K}^2 \tag{6}$$

where $\mathbf{I}$ is the identity matrix. When $t$ is the step for *Close*, the closed angle $c_t$ is computed as :

$$c_t = \text{Distance2Angle}\left(\frac{(s_t^{k+1} - P_t) \cdot \mathbf{V}_t}{\|\mathbf{V}_t\|^2}\mathbf{V}_t\right) \tag{7}$$

where $\text{Distance2Angle}$ is a function to map the cutting distance to the closed angle. The function is depended on the mechanical structure of scissors and we obtain it via real-world calibration.

## D   Details of the Benchmark

We consider the generation of curves in the uv space of the paper. In each track, curves are generated using Bézier curves parameterized by four control points, which are equidistant along the u-axis.

By connecting these four points with three line segments, we can compute the gradients of these lines. We approximate the second-order derivatives of the curve by calculating the differences in gradients between these lines. For the *Easy* track, the gradient differences are always either positive or negative, ensuring a consistent curve direction. For the *Middle* track, the gradient differences include both positive and negative values, creating more complex curves. Considering the robotic arm workspace, scissors cannot undergo significant rotations, i.e., exceeding $90°$, relative to the initial orientation along the moving trajectory. Empirically, we constrain the gradient of the first line to be within $[-\tan(40°), \tan(40°)]$ and the gradient of the last line within $[-\tan(60°), \tan(60°)]$.

# E Implementation of Policy Learning

## E.1 Point Cloud Pre-processing

We preprocess the visual input by cropping the global point cloud into a $3 \times 3 \times 3\,\text{cm}^3$ local patch centered around the scissors. This means that the policy only "sees" the local pattern rather than the entire curve. Since the local structures are generally consistent across curves, this approach helps the model generalize across different tracks.

## E.2 Vision Encoder

The sequential point cloud features from the PointNet++ encoder are concatenated and fed into three action heads. Each action head is a shallow MLP with the shape of $[L \times 512 - 256 - 64 - a]$, where $L$ is the length of the observation horizon and $a$ is the dimension of the action parameter. We train the model from scratch for 120,000 iterations. The learning rate is initialized at $1 \times 10^{-4}$ and decays by a factor of 0.1 at 60,000 and 110,000 iterations, respectively.

## E.3 Primitive Parameterization and Learning

The *push* primitive is parameterized by a 1D distance, indicating the amount of pushing along the current direction. The *rotate* primitive is parameterized using a rotation matrix, indicating the delta rotation in the scissors current frame. The *close* primitive is parameterized by a 1D distance, which represents the length it aims to cut. The *open* primitive is not parameterized separately because it shares a 1DoF with the close action. Essentially, we fix the open action to a predefined position (e.g., maximum opening) and do not need separate parameterization.

The model outputs parameters for all three primitives (rotate, close, push) in a single forward pass, but only selects one action to execute based on the predefined order and discards the other two outputs. The sequence of four action primitives is repeated until the curve is completed, which is referred to as **Recurrent Execution** in Fig. 2. For hyperparameter in the loss function, we simply choose $\lambda_p = \lambda_c = \lambda_R = 1$, and we find it is effective enough.

## E.4 Action Chunking

For Action Chunking, we follow the implementation of [7], which predicts $k$ future actions using an exponential weighting scheme where $w_i = e^{-m \times i}$, with $w_0$ representing the weight for the most recent action. In our experiments, we set $k = 4$ and $m = 0.01$.

# F More Discussion on Limitations

As we claim in the main paper, scissor cutting for non-simple or non-smooth curves is still an open problem. We adopt the definitions from MathWorld (https://mathworld.wolfram.com/). A curve is simple if it does not cross itself. A smooth curve is a continuous map from a one-dimensional space to an n-dimensional space that has continuous derivatives up to a desired order on its domain. Curves that do not meet these definitions are considered non-simple or non-smooth, respectively.

The limitation arises from the relatively long length of the scissors compared to regular end effectors. Due to the limited workspace, our small 6-DoF arm typically cannot rotate the scissors by more than 90 degrees while maintaining their position unchanged. As a result, non-smooth curves with significant turning angles are not achievable. Additionally, non-simple curves involve loops in the scissors' trajectory, which are also unattainable due to the constraints of the hardware setup, not the learning method itself. Addressing these limitations might require designing a dexterous dual-arm cooperation system, which could be a direction for future research.

