# OpenReview forum: "ScissorBot: Learning Generalizable Scissor Skill for Paper Cutting via Simulation, Imitation, and Sim2Real"
_robot-learning.org/CoRL/2024/Conference — CoRL 2024_

### Official Review · Reviewer_Wc2M · 2024-06-27

**Originality:** 4
**Technical Quality:** 4
**Clarity Of Presentation:** 4
**Potential Impact:** 3
**Recommendation:** 4
**Confidence:** 4

**Review:**

Pro
1. Novel simulation environment that accurately models paper cutting by adding vertices and splitting meshes according to the cutting trajectory.
2. Impressive sim2real capabilities on a dynamic task enabled by randomization methods, namely deviation correction by random rotation perturbations and simulating real-world edge bleeding in the depth observations.
3. Generalization and robustness of the policy to real-world perturbations caused by the complex dynamics of real-world paper cutting.
4. Clear presentation of the simulation design, impact of the randomization methods, and results.

Con
1. Lack of analysis of where the generalization capabilities of the policy stem from (see Questions For Rebuttal).
2. Subjectiveness of "success metric" and human comparison  (see Questions For Rebuttal).

Minor comments:

1. Equation 1 introduces λ_p, λ_c, λ_R but the paper never mentions what exact values were used for training.
2. Spelling mistakes: Line 19: achives -> achieves, Line 187: abaltion -> ablation

**Quality Of The Limitations Section:**

3

**Questions For Rebuttal:**

1. The paper provides insufficient information about the learned primitives. How are these parameterized? Why is Close parameterized but Open not? What does “Recurrent Execution” mean? Does the model predict parameters for all primitives in one forward pass or is there a conditioning of some form?

2. How much does the choice of λ_p, λ_c, λ_R (Equation 1) matter and how much tuning did you have to do?

3. How robust is the method to the type of paper and scissors used in the real experiments? What if the paperweight is increased, making it harder to cut? What if the scissors become dull over time? Or in other words: how much simulation tuning is involved to bridge the sim2real gap?

4. While the results show generalization capabilities of the trained policy from Easy to Middle and Hard tracks, it is unclear what design choice causes this. From an architecture and learning perspective, I’d assume the model would overfit to the Easy red lines seen in sim and perform poorly on unseen Medium and Hard tracks. The information provided is very vague and doesn’t provide sufficient explanations.
- Line 100-101 “The scissors are mounted on a single robotic arm, with the top edge of the paper fixed and the bottom edge free, enabling generalization to various scenarios.”
- Line 247-248 “This success is due to its generalizability from careful system design.”
A discussion of why the model generalizes would make the paper stronger.

5. Why is the oracle in Table 1 worse than the learned policy?

6. Line 253-254 “Finished rate, subjectively defining ”Finished” for trials which execute smoothly without falling into tearing, folding, or significant deviation from the target.”
As noted this metric is very subjective. While tearing and folding are intuitive, “significant deviation from the target” should be better defined. Instead, using a success metric based on thresholding the chamfer distance might provide more clarity.

7. Line 266-267 “[The] system achieves comparable performance to human single-hand manipulation under the same condition, which has only 2 mm error from the target curve.”
The paper provides no information on which setting the data was collected, how many subjects completed the study, and how proficient the subjects were at the task. Without a proper controlled use study, these numbers don’t mean anything.

**Robotics Focus:**

4

**Summary Of Paper:**

The presented work tackles the novel task of learning robotic scissor skills through a sim2real pipeline. The authors introduce a simulation for paper cutting that accurately models the behavior of the real world and use it to generate demonstrations. With a combination of vision-based imitation learning and randomization, the learned policies transfer to the real world and show generalization to unseen cutting shapes as well as robustness to perturbations of the paper.

**Summary Of Recommendation:**

The work aims to solve the novel task of robotic scissor skills, develops a corresponding simulation, and showcases the efficacy of imitation learning and sim2real to solve the task in a challenging real-world setting.

---

### Official Review · Reviewer_2CTS · 2024-07-20
**Solid experiments, novel idea, cool demo, limited scope**

**Originality:** 4
**Technical Quality:** 4
**Clarity Of Presentation:** 4
**Potential Impact:** 3
**Recommendation:** 3
**Confidence:** 4

**Review:**

The paper solves an interesting task: a robot learning to cut paper using a scissor.

Strengths:
- The task is very novel and interesting. Scissors are universal cutting tools. Learning to cut papers with scissors is more generalizable to different materials than specialized cutting machines.

- The paper proposes a novel thin-shell object simulation that looks very solid. This is traditionally very hard to simulate and I'm surprised that the learned skills from the simulation can be transferred to real with reasonable performance.

- The demo videos are impressive. It looks like the real-world experimental videos show that the proposed method can cut paper following the paths with a pretty high accuracy.

- The method, results, system setup, and figures are well-presented. I had no trouble understanding the method as well as the system setup. However, the authors should include more information regarding the hardware used in the system for reproducibility.

- I like the analysis experiments done in the paper, such as the effect of primitive learning.

Weaknesses:
- The first strength, however, is also a weakness of the paper. The authors didn't do a very good job demonstrating this point in the experiments. The paper itself focuses on cutting a flat normal-sized A4 paper which can be easily cut by a specialized machine. I think the author misses the opportunity here to show the learned technique can be used for various thin-shell materials such as tin foil or paper with various shapes, thicknesses, and strengths. Having these experiments will significantly help establishing the motivation why learning to use a generic tool -- scissor to cut paper is an important skill for robot to have.

- The system is very specialized towards scissor-paper-cutting. There are many tasks-specific method designs introduced in the system such as the special demonstration generation strategy.

- The hardest level of paths don't seem that hard. They are all smooth curves. See questions regarding this point.

**Quality Of The Limitations Section:**

1

**Questions For Rebuttal:**

- Can the lessons learned from this task be generalized towards other robotic tasks?
- Can the authors include some failure cases in the summary video as well as the main paper? What are some major failure modes?
- Is there cases where the scissor breaks the paper during pushing?
- What happens if the scissor cuts deep into the paper causing the base of the scissor or part of the robot arm collides with the paper causing it to deform in undesirable ways? Does this happen during experiments and will this influence the results?
- The authors mentioned that "Non-smooth curves like zig-zag patterns must result in the sudden relative pose change frequently for scissors while non-simple curves will result in a circle in the scissor trajectory. Both two kinds of scissors’ motion can’t be achieved considering our single-arm workspace." Could you clearly define what non-smooth and non-simple mean? Why are non-smooth patterns not achievable with a single-arm workspace? Isn't it the limitation of the method which doesn't allow sudden relative pose change instead of the single-arm workspace itself?

**Robotics Focus:**

4

**Summary Of Paper:**

The paper proposed a novel learning-based robotics system to use scissor to cut paper following predefined paths. The system is composed of a paper simulation engine, a learning method using 3D perception and primitive action, and a sim2real adjustment mechanism.

**Summary Of Recommendation:**

This paper presents a novel approach for a robot learning to cut paper with scissors, leveraging a robust thin-shell object simulation that transfers effectively to real-world applications. While the method and results are impressive, the scope of the paper is relatively limited. If the authors can better motivate the point why scissor is needed over specialized cutting machines, that'll make the paper's argument much stronger.

---

### Official Review · Reviewer_n7FW · 2024-07-22
**A well-designed system for paper cutting, but might be too restricted to this setting**

**Originality:** 3
**Technical Quality:** 3
**Clarity Of Presentation:** 3
**Potential Impact:** 2
**Recommendation:** 3
**Confidence:** 4

**Review:**

Strength:

The proposed task is quite challenging, and the results of this paper are promising. The authors also did a good job of describing the challenges of this task.

The authors propose a new simulator for paper cutting, which could be a valuable asset for the research community.

The paper proposes a well-engineered system to solve this task. The system combines imitation learning and sim-to-real, utilizing the advantages of both techniques.

Comprehensive evaluations of different components include the effect of primitive skills and comparisons to learning-based and classical methods.

Weakness:

I acknowledge the engineering efforts in tackling this complex task. However, I also notice this paper deals with a very specific problem. When evaluating the paper, I would more focus on whether the proposed technique could be extended to other similar scenarios. However, I’m concerned that the simulation, skill library, and the hardware setting are particularly designed for this single task. All these designs will be helpful for this problem but will be hard to extend to other scenarios. Thus, I’m worried that the work focuses too narrowly on a single task and lacks generalizability.

On the other hand, one way to demonstrate potential generalizability is to evaluate the system on other types of scissor skills. For example, cutting different materials (paper of different thicknesses, cloth, or fabric) would be a good demonstration of generalization. Using variations of scissors could also be a good demonstration. Conducting these demonstrations would highlight the paper's contribution from a skill learning perspective, rather than just showcasing a system for a very specific task.

**Quality Of The Limitations Section:**

3

**Questions For Rebuttal:**

From Figure 9(c), the target seems to be cutting along the red line instead of in the red line. Can you clarify how the goal is defined by the reward?

Can the authors comment on why previous deformable object simulators [10, 11, 12, 26] cannot simulate paper cutting? Is it due to efficiency issues or do they have fundamental limitations in terms of implementation?

Can the proposed system also generalize to different materials (such as rectangle cloth or fabric) or different types of scissors (with varying sharpness)?

How many trials when doing the evaluation?

**Robotics Focus:**

4

**Summary Of Paper:**

This paper proposes a complete solution for robotic paper cutting. The authors design a simulator, demonstration generation, and a sim-to-real learning pipeline to transfer the paper cutting skill to the real world. They first create a simulator for this task and generate demonstrations using heuristics given ground-truth information. They bridge the sim-to-real gap by introducing Deviation Correction and accurately modeling depth sensing noise. They also abstract the action space into a few primitive skills to simplify learning. The authors demonstrate improved performance over several competitive baselines.

**Summary Of Recommendation:**

This paper presents a well-engineered effort for a complicated problem. I’m concerned with the generalizability of the proposed system.

---

### Author Rebuttal · Authors · 2024-08-11

We include a new video material to support the discussion with AC and reviewers. The main content of the video is as follows:
+ Demonstrations for cutting different materials
+ A common failure case during cutting

---

### Decision · Program_Chairs · 2024-09-05

**Decision:**

Accept

**Comment:**

This paper proposes a pipeline for scissor cutting. The method is somewhat domain-specific, but results are good enough for this to be a valuable contribution to the robot learning community. The rebuttal led to two reviewers increasing their scores, so now there is a strong accept and two weak accepts. The authors are encouraged to add more details about the user study to the final paper.

Strengths:

* A novel pipeline for sim2real cutting, with real world experiments shown.
* The cutting application is distinct from prior work that has done cutting.

Weaknesses:

* Multiple reviewers commented that the application might be too narrow (cutting). It would be good to motivate the need for this over specialized cutting machines.
* The success metric might be too subjective, and the paper should clarify the comparison with human performance.